

**New observations of the distribution, morphology, and dissolution dynamics of**
**cryogenic gypsum in the Arctic Ocean**
**Jutta E. Wollenburg[1*], Morten Iversen[1,2], Christian Katlein[1], Thomas Krumpen[1], Marcel**
**Nicolaus[1], Giulia Castellani[1], Ilka Peeken[1], Hauke Flores[1]**
[1]Alfred-Wegener-Institut Helmholtz-Zentrum für Polar- und Meeresforschung, D-27570,
Bremerhaven, Germany
[2]MARUM and University of Bremen, D-27359, Bremen, Germany
[*]Corresponding author and requests for materials should be addressed to J.E.W. (email:
Jutta.Wollenburg@awi.de)



## Abstract

To date observations on a single location indicate that cryogenic gypsum ($Ca[SO_4] \cdot 2H_2O$) may constitute an efficient but hitherto overlooked ballasting mineral enhancing the efficiency of the biological carbon pump in the Arctic Ocean. In June-July 2017 we sampled cryogenic gypsum under pack-ice in the Nansen Basin north of Svalbard using a plankton net mounted on a Remotely Operated Vehicle (ROVnet). Cryogenic gypsum crystals were present at all sampled stations, which suggested a persisting cryogenic gypsum release from melting sea ice throughout the investigated area. This was supported by a sea ice backtracking model that indicated that gypsum release was not related to a specific region of sea ice formation. The observed cryogenic gypsum crystals exhibited a large variability in morphology and size, with the largest crystals exceeding a length of 1 cm. Preservation, temperature and pressure laboratory studies revealed that gypsum dissolution rates accelerated with increasing temperature and pressure, ranging from 6% $d^{-1}$ by mass in Polar Surface Water (-0.5 °C) to 81% $d^{-1}$ by mass in Atlantic Water (2.5 °C at 65 bar). When testing the preservation of gypsum in Formaldehyde-fixed samples we observed immediate dissolution. Dissolution at warmer temperatures and through inappropriate preservation media may thus explain why cryogenic gypsum was not observed in scientific samples previously. Direct measurements of gypsum crystal sinking velocities ranged between 200 and 7000 m $d^{-1}$ indicated that gypsum-loaded marine aggregates could rapidly sink from the surface to abyssal depths, supporting the hypothesised potential as a ballasting mineral in the Arctic Ocean.

## Keywords:

Cryogenic gypsum, Arctic Ocean, mineral ballasting. biological carbon pump, sea ice.



## 1    Introduction


Climate change in the Arctic Ocean has led to a drastic reduction of summer sea ice extent as
well as to a significant thinning of the sea ice (Kwok, 2018; Kwok and Rothrock, 2009). Sea
ice strength has reduced, and increased deformation and fractionation result in a progressively
increasing sea ice drift speed (Docquier et al., 2017) and sea ice export. Over the past decades
the ice export via the Fram Strait alone has increased by 6% and 11% per decade as annual
mean, and during the productive spring and summer period, respectively (Smedsrud et al.,
2017). An increasing amount of sea ice produced in the East Siberian and Laptev Sea melts
over the adjacent continental slopes or in the central Arctic Ocean (Krumpen et al., 2019).
Overall, the Arctic Ocean sea ice cover has shifted to a predominantly seasonal ice cover.
However, although the majority of sea ice diminishes during late summer, the amount of sea
ice produced in autumn to winter progressively increases (Kwok, 2018).
Large-scale transformations in the seasonal sea ice cover impact the physical, chemical and
biological dynamics of the sea ice-ocean system. However, especially the interactions of
physical-chemical processes within the sea ice and pelagic to benthic biological processes
have only received little attention. Of particular importance are poorly soluble minerals
precipitated within the brine channels of sea ice which, once released, may ballast organic
material sinking to the sea-floor. The changing icescape with more leads and the thinner
Arctic sea ice allows increasing light penetration into the under-ice surface water (Katlein et
al., 2015; Nicolaus et al., 2013; Nicolaus et al., 2012), supporting fast-growing and often
massive under-ice phytoplankton blooms (Arrigo et al., 2012; Arrigo et al., 2014; Assmy et
al., 2017). A recent study reported on a sudden export event of an under-ice bloom of the
'unsinkable alga' *Phaeocystis,* caused by the ballasting effect of cryogenic gypsum released
from melting sea ice (Wollenburg et al., 2018a). This single event was the first and only
report of cryogenic gypsum release in the Arctic Ocean. Moreover, this sea ice precipitation
of cryogenic gypsum has never been recorded in Arctic sediments, sediment traps or other
field studies.
When sea ice forms, the concentrations of dissolved ions in brine increase, and depending on
the temperature of sea ice, a series of minerals (ikaite, mirabilite, hydrohalite, gypsum,
hydrohalite, sylvite, $MgCl_2$, Antarcticite) precipitate (Butler, 2016; Butler and Kennedy, 2015;
Geilfus et al., 2013; Golden et al., 1998; Wollenburg et al., 2018a). Once released into the
ocean, gypsum seems to be the most stable of the cryogenic precipitates (Butler et al., 2017;



Strunz and Nickel, 2001). Sea ice derived cryogenic gypsum was firstly described by Geilfus
et al. (Geilfus et al., 2013), in a comprehensive work on the chemical, physical, and
mineralogical aspects of its precipitation in experimental and natural sea ice off Greenland.
According to FREZCHEM, a chemical–thermodynamic model that was developed to quantify
aqueous electrolyte properties at sub-zero temperatures, cryogenic gypsum can precipitate at
temperatures below –18 °C, and within a small temperature window between –6.5 and –8.5
°C (Geilfus et al., 2013; Marion et al., 2010; Wollenburg et al., 2018a). However,
measurements on the stoichiometric solubility products showed that gypsum dynamics in ice–
brine equilibrium systems strongly depend on the solubility and precipitation of hydrohalite
and mirabilite (Butler, 2016; Butler et al., 2017). So far gypsum precipitation in experimental
setups were only observed at temperatures between -7.1 and -8.2 °C, and not in the lower
temperature range (Butler, 2016; Butler et al., 2017). Moreover, as Arctic sea ice rarely
reaches temperatures lower than -18 °C, cryogenic gypsum is more likely precipitated within
the higher temperature window in the Arctic Ocean (Wollenburg et al., 2018a).
A model applied to understand the gypsum release event of 2015 showed that the ice flow
was to warm when it started to form and identified December to February as the most likely
time span for gypsum precipitation (Wollenburg et al., 2018a). Due to the absence of a
downward brine flux in this advanced phase of sea ice formation, gypsum crystals likely
remain trapped in the ice until spring. In the absence of sufficient field observations gypsum
release from sea ice is expected to peak at the beginning of the melting season, when sea ice
warms to temperatures above -5 °C. This temperature marks the transition in the fluid
transport capacities of sea ice allowing brine water and included crystals to be released into
the water column (Golden et al., 1998). However, in lack of any extensive, best year-round
field studies our knowledge depends on models, kinetics and two single field observations
(Geilfus et al., 2013; Wollenburg et al., 2018a). There are no studies on sea ice-derived
cryogenic gypsum crystal morphologies and its stability in seawater. It is unclear whether
gypsum just precipitates during the assumed peak in December to February or whether it
continues to grow from remaining brines during sea ice drift. We therefore need more studies
on the formation and release of cryogenic gypsum to assess its impact on biogeochemistry in
the Arctic and sub-Arctic.
In this study, we systematically investigated the occurrence of cryogenic gypsum release from
sea ice in spring 2017 with special emphasis on the overall appearance of the crystals.
Varieties of cryogenic gypsum crystal morphologies are described and illustrated. The



sampled gypsum crystals were further subjected to various laboratory experiments. Hereby,
the dissolution behaviour over typical depth- and temperature ranges of the Arctic water
column, in Formaldehyde solution typically used in biological sampling were investigated and
the sinking speed of gypsum crystal measured. These experiments were conducted to answer
the question, why cryogenic gypsum has not previously been observed in field studies and if it
qualifies as ballast mineral.


**2      Material and Methods**
**2.1      Gypsum sampling with the ROVnet and on-board treatment**
RV *Polarstern* expedition PS 106 (June-July 2017) in the early melting season gave the
opportunity to systematically study the occurrence of cryogenic gypsum release and the
overall appearance of crystals in the area north of the Svalbard and on the Barents Sea shelf
(Fig. 1A; Table 1).
Cryogenic gypsum was sampled from the upper 10 m of the under-ice water at four stations
distributed throughout the expedition area (Fig. 1A; Table 1). The first part of the expedition
(PS106/1) consisted of a drift study to the north of Svalbard, during which the vessel was
anchored to an ice floe (station 32). This ice floe was revisited 6 weeks later at the end of the
expedition (PS106/2) (station 80). During the second part of the expedition (PS106/2),
cryogenic gypsum was collected over the western Barents Sea (station 45) and in the Nansen
Basin to the north-east of Svalbard (station 66). Gypsum crystals were sampled with a
plankton net mounted on a remotely operated vehicle (ROVnet). The ROVnet consists of a
Polycarbonate frame with an opening of 40 cm by 60 cm, to which a zooplankton net with a
mesh size of 500 $\mu$m was attached (Flores, 2018). For gypsum sampling, a handmade nylon
net with an opening of 10 cm by 15 cm and a mesh size of 30 $\mu$m was mounted in the
zooplankton net opening. The concentrated particulate material of the small nylon net was
collected in a 2 L polyethylene bottle attached to the cod end of the net. A gauze-covered
window in the cod-end bottle allowed seawater to drain off. Both nets were mounted on the
aft end of a M500 (Ocean Modules, Sweden) observation class ROV carrying an extensive
sensor suite described in Katlein et al. (Katlein et al., 2017). After each ROVnet deployment,



the nets were rinsed with ambient sea-water to concentrate the sample in the cod end of the
net. The ROVnet sampled horizontal profiles in the water directly below the sea ice. Standard
ROVnet profiles were conducted at the ice-water interface, at 5 m and at 10 m depth. The
distance covered by each profile ranged between 300 and 600 m. At station 32, the 10 m
profile was aborted due to technical failure, and at station 80 no 5 m profile was sampled due
to time constraints, and the subsurface sample was discarded due to handling failure (Table

137 1).

The concentrated particulate material collected in the cod-end bottle of the gypsum sampling
net was mixed with a sample equivalent volume of 98% ethanol, and stored at 4 °C until
further analyses (Wollenburg et al., 2018a).
At ROVnet sampling stations, ice thickness was estimated through thickness drill holes with a
tape measure. To characterize the properties of the ice floes sampled on the floe-wide scale,
ice thickness surveys were conducted at each sampling station with a GEM2 (Geophex)
electromagnetic induction ice-thickness sensor (Katlein et al., 2018).
**2.2    Initial analyses of ROVnet samples**
In the home laboratory the samples were rinsed onto a 32 $\mu$m mesh using fresh water. The
samples were then oven-dried at 50°C for 20 hours. The remaining crystals were transferred
into pre-weighed micropaleontological slides, and their weight was determined with a high-
precision Sartorius SE2 ultra-microbalance. Under a Zeiss Axio Zoom V16 microscope,
pictures were taken with an Axiocam 506 colour camera. We made both overview images of
the whole sample and detailed images of individual crystals. From all samples and crystal
morphologies, individual crystals were analysed using Raman microscopy, which confirmed
that the crystals were gypsum (Wollenburg et al., 2018a). As in some samples both, very large
and very small crystals (Figs. S3-S4) were observed, the >32 $\mu$m samples were dry-sieved
over a 63 $\mu$m analysis sieve. The length and width of the cryogenic gypsum crystals in the
size fractions >32<63 $\mu$m and >63 $\mu$m was determined with the software application ImageJ
on 50 crystals in each sample and size fraction (Schneider et al., 2012) (Tab. 2).
**2.3    Initial analyses of ice cores**
At all ice stations, sea ice cores for archive purposes and for further measurement of bottom
communities were drilled with a 9 cm diameter ice corer (Kovacs Enterprise) and stored at -


20°C (Peeken, 2018). One ice-core from station 80 and four bottom slices (10 cm) of ice-
cores from station 45 were studied to investigate the gypsum crystal morphologies within sea
ice. Each section was transferred into a measuring jug with lukewarm tap water for approx.
two seconds, and then the jug was emptied over a 32 $\mu$m analysis sieve, and repeatedly
refilled. This process was continued until all ice was melted. With the aid of a hand shower
and a wash bottle the residue on the sieve was rinsed and transferred into a 30 $\mu$m mesh-
covered funnel, dried and transferred into a micropaleontological picking tray for inspection
and documentation. For storage, the residue was transferred into pre-weighed labelled
micropaleontological slides.

## 170    2.4      Dissolution experiments

The aim of our dissolution experiments was to investigate the persistence of gypsum crystals
against dissolution in the Arctic water column (water mass trials) and under common
biological sample treatment (Formaldehyde trial).
Dissolution experiments were carried out on individual gypsum crystals collected from
ROVnet samples. Hereby, 5 cryogenic gypsum crystals with different crystal morphologies,
and from both size fractions were used in each reaction chamber. Before the start and after the
termination of each experiment, pictures of the cryogenic gypsum crystals used were taken
with an Axiocam 506 colour camera under a Zeiss Axio Zoom V16 microscope. The weight
of the crystals before and after each treatment was determined with a high-precision Sartorius
SE2 ultra-microbalance after they had been transferred into a pre-weighted silver boat. The
experimental running time of each experiment was 24 hours.

### 182    2.4.1    Water mass trials

The experiments to simulate dissolution within the different water masses and hydrostatic
pressure regimes of the Arctic Ocean were carried out with high-pressure chambers installed
in a cooling table (Wollenburg et al., 2018b). With a high-pressure pump (ProStar218 Agilent
Technologies), peak tubing, and multiple titanium valves a continuous isobaric and isocratic
one-way seawater flow of 0.3 ml/min was directed through a set of four serially arranged
high-pressure chambers each with an internal volume of 0.258 ml (Wollenburg et al., 2018b).
This setup allowed for dissolution experiments at defined pressures and temperatures
(Wollenburg et al., 2018b). For the experiments, we used sterile-filtered (0.2 $\mu$m mesh) North
Sea water that was adjusted to a salinity of 34.98 by addition of 1 g Instant Ocean® sea salt



per L and psu-offset. The natural pH of 8.1 after equilibration to the refrigerator's atmosphere
(at 2.5 °C and at atmospheric pressure), lowers to pH 8.05 at 2.5 °C at 150 bar (Culberson and
Pytkowicx, 1968). Five experiments, with 4 high-pressure chambers were carried out. The
Polar Surface (PSW) water corresponding experimental trial was running at -0.5 °C and 3 bar,
the experimental Atlantic Water (AW) trial at +2.5 °C and 65 bar, and three experimental
Deep Water trials were conducted at -1 °C and 100, 120 and 150 bar, respectively.
**2.4.2    Formaldehyde trial**
To study the effect of Formaldehyde treatment on cryogenic gypsum, the crystals were
subjected to a Formaldehyde solution of 4% in seawater, which is commonly used to preserve
biological samples. The stock solution consisted of 500 ml Formaldehyde concentration of
40%, 500 ml aqua dest. and 100 g hexamethylenetetramine, adjusted to a pH of 7.3-7.9.
Aliquots of the 20% stock solution were added to the four-fold volume of artificial Arctic
Ocean sea water to obtain a final concentration of 4%.
The Gypsum crystals were transferred into Falcon Tubes, and the 4% Formaldehyde solution
was added. The Falcon tubes were then either stored at 3 °C, or at room temperature. After
the experiments, the gypsum crystal-Formaldehyde suspension was washed with deionized
water over a 10 $\mu$m mesh using a wash bottle, and dried on gauze. As in all formaldehyde
trials all gypsum dissolved, no post-experimental weight was determined.

**2.5    Size-specific settling velocities of gypsum**

The size-specific sinking velocity of cryogenic gypsum was measured in a settling cylinder
(Ploug et al., 2008). The cylinder (30 cm high and 5 cm in diameter) was filled with filtered
seawater (salinity 32) and surrounded by a water jacket for thermal stabilization at 2 °C. The
settling cylinder was closed at both ends, only allowing insertion of a wide-bore pipette at the
top. Immediately before measurement, the gypsum was submerged into seawater with a
salinity of 32 and a temperature of 2 °C, and then transferred to the settling cylinder with a
wide-bore pipette. The gypsum crystals were allowed to sink out of the wide-bore pipette,
which was centered in the cylinder. The descent of the pellets was recorded by a Basler 4
MPixel Ethernet camera equipped with a 25 mm fixed focal lens (Edmund Optics). The
settling column was illuminated from the sides by a custom-made LED light source. The
camera recorded 7 images per second as the gypsum crystals sank through the settling
column. The setup was calibrated by recording a length scale before sinking velocity
measurements. The size and settling of the individual gypsum crystals was determined with



the image analysis software ImageJ. This was done by using the projected area of the crystals
to calculate the equivalent spherical diameter and the distance traveled between the
subsequent images to determine the sinking velocity of the individual crystals (Iversen et al.,

229 2010)

We calculated the excess density ($\Delta\rho$) ($\Delta\rho$ = gypsum density – water density) of the crystal
from the Stokes drag equation:

$$\Delta\rho = \frac{C_D \rho_w SV^2}{\frac{4}{3}gESD} \tag{1}$$


where $C_D$ is the dimensionless drag force (equation 2), $\rho_w$ is the density of seawater (1.0256 g
cm³, for a salinity of 32 at 2 °C), $SV$ is the measured sinking velocity in cm s¹, $g$ is the
gravitational acceleration of 981 cm s², and $ESD$ is the equivalent spherical diameter in cm.
We calculated $C_D$ using the drag equation for low Reynolds numbers (White, 1974):

$$C_D = \left(\frac{24}{Re}\right) + \left(\frac{6}{1+Re^{0.5}}\right) + 0.4 \tag{2}$$


where the Reynolds number (Re) was defined as

$$Re = SV\ ESD\ \frac{\rho_w}{\eta} \tag{3}$$

where $\eta$ is the dynamic viscosity (1.7545 × 10² g cm¹ s¹ for a salinity of 32 at 2 °C). Equation
2 is valid up to a Reynolds number of 2x10^5 (Vogel and Beety, 1994). The gypsum crystals
had Reynolds numbers ranging from 0.77 to 128.

**2.6      Backtracking the sampled ice flows under which cryogenic gypsum was sampled**

To determine sea ice drift trajectories of sampled sea ice we used a Lagrangian approach
(IceTrack) that traces sea ice backward or forward in time using a combination of satellite-
derived low resolution drift products. So far, IceTrack has been used in a number of
publications to examine sea ice sources, pathways, thickness changes and atmospheric
processes acting on the ice cover (Damm et al., 2018{Peeken, 2018 #13678; Krumpen et al.,
2016; Peeken et al., 2018). A detailed description is provided in Krumpen et al. (Krumpen et
al., 2019).

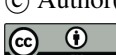



Sea ice motion information was provided by different institutions, obtained from different
sensors, and for different time intervals. In this study we applied a combination of three
different products: (i) motion estimates based on a combination of scatterometer and
radiometer data provided by the Center for Satellite Exploitation and Research (CERSAT
(Girard-Ardhuin and Ezraty, 2012), (ii) the OSI-405-c motion product from the Ocean and
Sea Ice Satellite Application Facility (OSISAF (Lavergne, 2016), and (iii) Polar Pathfinder
Daily Motion Vectors from the National Snow and Ice Data Center (NSIDC (Tschudi et al.,

266 2016).

The tracking approach works as follows: An ice parcel is traced backward or forward in time
on a daily basis. Tracking is stopped if a) ice hits the coastline or fast ice edge, or b) ice
concentration at a specific location drops below 50% and we assume the ice to be formed.
The applied sea ice concentration product was provided by CERSAT and was based on
85 GHz SSM/I brightness temperatures, using the ARTIST Sea Ice (ASI) algorithm.

**3  Results**
**3.1  Presence and distribution of cryogenic gypsum under the investigated ice-floes**
Based on backtracking (Krumpen, 2018) and sea ice observations, the sampled ice-floes had
an age of 1 to 3 years (Fig. 1B) and were originating from the Siberian Sea (station 32/80), the
Laptev Sea (station 45), and were more locally grown in the Nansen Basin (station 66).
Whereas the mean sea ice thickness at the ROV survey stations ranged between 94 and 156
cm, the mean sea ice thickness of the investigated ice-floes estimated by an ice-thickness
sensor surveys (Katlein et al., 2018) was 1.90 m for station 32, 1.00 m for station 45, and 1.80
m for stations 66 and 80 (Fig. 1A, Table 1). Despite the different origins and thicknesses of
sea ice, cryogenic gypsum crystals were found at all stations and in all depth layers sampled
with the ROVnet (Figs. 1A, B, Tab. 1). At all stations and sampling depths the samples were
dominated by cryogenic gypsum, with a proportional dry weight of >96.5% in the 5 m-sample
at station 32, and with >99% in all other samples (Figs. 2, Figs. S1-S4). Other lithogenic
particles, as often found in sea ice (Nürnberg et al., 1994), were essentially absent.
**3.2  The morphology of cryogenic gypsum**

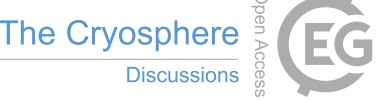

The samples collected at station 32 were dominated by rounded, matte, solid cryogenic
gypsum crystals with a mean length-width ratio of 1.40-1.76 (Tab. 2, S1). The proportional
mass contribution of the smaller-sized crystals of the >30<63 $\mu$m size fraction increased with
depth and outweighed the contribution of the >63 $\mu$m size fraction with 56.30%, and 66.28%
for the 0 and 5 m water depth sample, respectively (Fig. 3). At 0 m, the mean length of the
crystals was 68.46 $\mu$m in the >63 $\mu$m size fraction and 44.27 $\mu$m in the >30<63 $\mu$m fraction.
At 5 m depth, crystal dimensions were similar, ranging at mean crystal lengths of 63.28 $\mu$m in
the >63 $\mu$m, and 35.90 $\mu$m in the >30<63 $\mu$m size fraction, respectively.
At station 45, the crystals were mostly solid and for most part hyaline, rather than matte
crystals as at station 32 (Figs. 2C-D, 6, S2). With decreasing weight proportion, the >63 $\mu$m
size clearly dominated the 0, 5, and 10 m samples with 79.90, 73.39, and 66.14%,
respectively. In the 0 m layer samples, mean crystal lengths were 114.18 $\mu$m in the >63 $\mu$m
size fraction and 58.74 $\mu$m in the >30<63 $\mu$m size fraction (Tab. 2). At 5 m depth, we
observed mean crystal lengths of 111 $\mu$m in the >63 $\mu$m size fractions, and 56.73 $\mu$m in the
>30<63 $\mu$m fraction. The mean crystal lengths in the 10 m sample was 92.83 and 50.32 $\mu$m
for the >63 and >30<63 $\mu$m size fraction, respectively. At station 45 the crystal length-width
ratio varied between 1.37 and 1.98, measured in the >30<63 $\mu$m size fraction of the surface
sample, and the >63 $\mu$m size fraction of the 10 m sample. The cryogenic gypsum crystals
retrieved from the melted ice core drilled at this station were solid and hyaline. In size and
shape they resembled the crystals of the 10 m layer at this station, with a mean crystal length
of 114.2 $\mu$m, mean width of 57.2 $\mu$m, and a length-width ratio of 2 (Fig. 4).
At station 66, the crystals from 0 m water depth were dominated by large, pencil-like, hyaline
and solid crystals with a mean crystal length of 1,355 $\mu$m and mean width of 415 $\mu$m in the
dominating >63 $\mu$m fraction (99.25% mass) (Fig. 2B, S3, Tab. 2). These crystals with an
average length-width ratio of 3.27 were found as isolated crystals, but very often also as inter-
grown crystal rosettes with two to more than 10 individual crystals involved (Fig. S3; Tab. 2).
The >30<63 $\mu$m size fraction (0.75% mass) was dominated by matte, whitish, rounded
gypsum particles and tiny gypsum needles with a mean crystal length of 56.67 $\mu$m (Fig. S3,
Tab. 2.). As at the other stations the weight proportion of the >63 $\mu$m size fraction
significantly decreased from 99.25 in the 0 m, to 75.23 at 5 m, and 61.18% in the 10 m
sample (Fig. 2). The size of cryogenic gypsum crystals collected from the 5 and 10 m layers
was significantly smaller and predominantly composed of isolated small hyaline and euhedral
gypsum needles. The length-width ratio ranged between 5.60 (5 m) and 4.37 (10 m) (Figs.





2A, S3, Tab. 2). In the 5 m layer sample, the mean crystal length was 411.42 $\mu$m in the >63
$\mu$m size fraction, and 62.03 $\mu$m in the >30<63 $\mu$m size fraction. The 10 m samples showed a
mean crystal length of 101.40 $\mu$m in the >63, and 30.71 $\mu$m in the >30<63 $\mu$m size fraction
(Tab. 2).
In the 10 m layer sample of station 80, large tabular gypsum crystals measuring up to 1 cm in
length (mean length: 3,078 $\mu$m, mean width: 1,830 $\mu$m) dominated the >63 $\mu$m size fraction.
Their average length-width ratio was 1.7. This size fraction contributed 89.1% of the gypsum
mass (Figs. 5, S4, Tab. 2). The >30<63 $\mu$m size fraction was composed of fragments of these
large crystals and few small gypsum needles. These often intergrown columnar crystals
looked bladed, for most part also dented and with numerous cracks. Their mean length was
71.8 $\mu$m. The ice core retrieved from this station was very porous and broke into pieces of 9
to 11 cm. Cryogenic gypsum was retrieved from all these ice core sections and revealed a
dominance of extraordinary large crystals (Figs. 5, S4), resembling the ROVnet samples from
this station. The largest cryogenic gypsum crystals >6,000 $\mu$m (mean crystal length: 2,821
$\mu$m, mean width: 1,689 m) were retrieved from the top-most 8 cm ice core section, whereas,
the maximum crystal size gradually decreased downcore (Fig. S4). The crystals themselves
lacked sharp corners, and the large crystals had cavities inside, indicating an advanced stage
of dissolution (Figs. 5C-D; S4).

### 3.3    Dissolution experiments

#### 3.3.1    Experiments to simulate cryogenic gypsum dissolution within the Arctic water column

Our study area was characterized by the presence of three main water masses (Nikolopoulos
et al., 2018; Rudels, 2015): 1) The Polar Surface Water (PSW) including the halocline, with a
variable mean salinity of 32 and a temperature range of -1.8 to 0.0 °C, extended from the
surface to maximum 100 m water depth (Nikolopoulos et al., 2018). 2) The Atlantic Water
(AW) with a mean salinity of 34.4 to 34.7 and variable temperature of 0.0 to 4.7 °C in the
study area extended from below the PSW to 600-800 m water depth (Nikolopoulos et al.,
2018). 3) The Eurasian Arctic Deep Water (EADW) fills the deep Eurasian Basin below the
AW with a temperature range of <0 to -0.94 °C and a salinity of about 34.9 (Nikolopoulos et
al., 2018).



The dissolution experiments carried out to simulate dissolution in the PSW were set to 3 bar,
-0.5 °C. Over the 24 hours lasting PSW-simulating dissolution experiment, about 6% of the
gypsum dissolved (Figs. 6, S5A, Tab. 3). In the AW experiment, the combination of positive
temperatures (2.5 °C) and a pressure of 65 bar impacted the dissolution on the cryogenic
gypsum crystals more than in any other seawater trial. More than 80% of the cryogenic
gypsum crystals dissolved during the 24-hours experiment (Figs. 6, S5B, Tab. 3). Moreover,
as dissolution mainly affects the crystal's surface, smaller gypsums crystals and those with
increased surface roughness were preferentially impacted by dissolution, whereas larger and
solid crystals with smooth surface showed the lowest dissolution (Fig. S5B). The EADW-
simulating dissolution experiments set to a temperature of -0.5 °C showed a progressive
cryogenic gypsum dissolution of 26, 58, and 62% with increasing pressure for the 100, 120
and 150 bar experiments, respectively (Figs. 6, S6, Tab. 3).
**3.3.2   Experiments to simulate cryogenic gypsum dissolution within Formaldehyde-**
**treated biological samples**
In the Formaldehyde experiments we exposed our set of cryogenic gypsum crystals to a
Formaldehyde solution of 4%, which is commonly used to store pelagic samples from the
Polar Oceans (Edler, 1979). Irrespective of the temperature at which the sample was stored,
all gypsum dissolved within 24 hours.
**3.4     Sinking velocities of gypsum crystals**

The sinking velocity (*SV*) of the gypsum crystals increased with crystal size (Fig. 7A). Small
crystals with an equivalent spherical diameter (*ESD*) of 200 $\mu$m sank with 300 m d$^{-1}$ while
large gypsum crystals with ESDs of 2,000 to 2,500 $\mu$m sank with velocities of 5,000 to 7,000
m d$^{-1}$. The size to settling relationship was best described by a power function (SV = 4239.9
ESD$^{0.39}$, R$^2$ = 0.84). As the power function suggests, the settling velocity levelled off for the
largest gypsum crystals (Fig. 7A). The observed excess density of all crystals was smaller
than is expected from the density of gypsum (2310 kg/m$^3$). For the visually non porous
smaller crystals drag, the deviation of gypsum crystals from round particles, and dissolution
may be the main reason for the calculated lower density.
However, plotting the excess density as a function of size (Fig. 7B) also showed that the
excess density of the gypsum decreased with increasing crystal size. The microscopic images
show that large crystals were more porous and had more complex shapes (Fig. S8 A-C)



compared to the small crystals that were more spherical and less porous (Figs. 2, 4-5, S8 D).
Hence, the flat settling to size relationship for large gypsum crystals (Fig. 7A), was essentially
due to a combination of increased porosity causing decreasing excess density and increased
drag due to the complex shapes of the large crystals.

## 4      Discussion



### 4.1      Distribution and morphology of cryogenic gypsum crystals



This study shows for the first time the wide-spread presence of cryogenic gypsum under melting
Arctic sea ice of different origin. At all stations cryogenic gypsum dominated the sample
fraction of particles >30 $\mu$m in Eurasian Basin surface waters, indicating a continuous cryogenic
gypsum flux from warming sea ice over a period of six weeks.
When designing the ROVnet for cryogenic gypsum sampling, we opted for the coarser >30 $\mu$m
mesh to prohibit an overflow of the sampling container when running into a phytoplankton
bloom. However, as Geilfus et al. (Geilfus et al., 2013) had observed gypsum crystals as small
as 10 $\mu$m, we probably lost an unknown proportion of smaller gypsum crystals by the chosen
sampling strategy. The gypsum crystals described from sea ice so far retrieved from only 3-
days-old experimental and 30 cm thick natural sea ice off Greenland were small (crystal length
max. 100 $\mu$m), planar euhedral gypsum crystals often intergrown or as rosettes (Geilfus et al.,
2013). Similar, but larger (crystal length up to 1 mm), gypsum crystals were observed within
*Phaeocystis* aggregates collected in the region of the present study (Wollenburg et al., 2018a).
However, here we show that gypsum crystals exhibit a strong variability in size and
morphology. Particularly large crystals were characterised by more complex shapes (Fig. 2, 5,
S3-4) and increased porosity (Figs. S6A-C), compared to the small planar euhedral (Fig. 2A)
and more spherical crystals (Fig. S6D). Euhedral crystal needles larger but otherwise similar to
those described by Geilfus et al. (Geilfus et al., 2013) and Wollenburg et al. (Wollenburg et al.,
2018a) dominated the >63 $\mu$m fraction collected at 5 and 10 m depths at station 66, and smaller
crystals contributed especially to the >30<63 $\mu$m size fraction of the station's subsurface
samples.
As cryogenic gypsum forms in sea ice brine pockets or channels, the size and morphology
especially of large crystals is likely determined by sea ice texture and porosity during gypsum



precipitation. Pursuing this hypothesis, the large and intergrown crystals collected from the
0 m layer at station 66, and the 10 m layer and ice-core at station 80, formed in highly
branched granular sea ice (Lieb-Lappen et al., 2017; Weissenberger et al., 1992). In contrast,
the small cryogenic gypsum needles reported by Geilfus et al. (Geilfus et al., 2013) and
Wollenburg et al. (Wollenburg et al., 2018a), may have preferentially formed in columnar sea
ice. Even sampling the same ice-floe (station 32 and 80), the appearance of the crystals
changed. Possibly, a widening of the brine channels during the elapsed time (6 weeks)
allowed a release of larger crystals at station 80 when compared to station 32. However,
crystal growth during this elapsed period or lateral advection of large crystals cannot be
excluded. Thus, detailed texture analyses on sea ice cores prior to sampling are needed to
validate or reject hypotheses on a link between sea ice porosity and cryogenic gypsum crystal
size and morphology and should be considered in future studies.
The sea ice microstructure dictating the formation of gypsum crystals in the brine matrix
likely varied among ice-floes due to different ages, origins and drift trajectories (Fig. 1B). For
example, station 66 was the only station where the sea ice likely formed over the central
Nansen Basin only months before our study (Fig. 1B). The surface sample of station 66 had
large intergrown hyaline star-shaped gypsum crystals that were observed at no other station.
They also showed a considerably higher length-width ratio than crystals from second-year ice
of stations 32/80 and 45 (Fig. 1B; Fig. 2). Accordingly, a close relationship between local sea
ice properties and gypsum crystal morphology in the underlying water was evident from the
comparison of gypsum crystals collected with the ROVnet with those retrieved from ice cores
collected at two stations. The ice-core samples revealed cryogenic gypsum crystals that
basically resembled the crystal morphologies collected from the water column at these
stations, indicating that the gypsum morphologies observed in the water column likely reflect
the gypsum precipitation conditions and brine-channel structure of local ice-floes. The current
understanding of mineral precipitation in supersaturated brines relies on ice-core analyses, sea
ice brine- and experimental studies, and on mathematical modelling of the temperature
window in which each mineral is likely to form (Butler et al., 2017; Marion et al., 2010).
There are still many uncertainties regarding the precipitation and dissolution of gypsum
within natural sea ice and during ice-core storage. Although the FREZCHEM model and
Gitterman Pathway predict gypsum precipitation under defined conditions, only Geilfus et al.
(Geilfus et al., 2013) and Butler et al. (Butler et al., 2017) succeeded in retrieving gypsum
under such conditions, whereas others failed (Butler and Kennedy, 2015). According to the
FREZCHEM model, cryogenic gypsum precipitates at temperatures of -6.2 to -8.5 °C and at



temperatures <-18 °C (Geilfus et al., 2013; Wollenburg et al., 2018a). Accordingly, a storage
temperature of -20 °C would allow the post-coring precipitation of gypsum from contained
brines. However, in field and experimental studies cryogenic gypsum was so far only
observed to precipitate in the -6.2 to -8.5 °C temperature window, even when treatments were
conducted below -20 °C  (Butler et al., 2017; Geilfus et al., 2013). Furthermore, the observed
signs of dissolution on the large cryogenic gypsum crystals from the ice-core when compared
to the sharp-edged crystals retrieved from the water column at station 80 indicate that
significant new precipitation of gypsum during storage did not occur, rather the opposite.
Apart from the growing conditions of gypsum crystals within sea ice, the size spectrum of
crystals retrieved from different depths in the water column likely was essentially altered by
the size-dependent sinking velocity of the crystals. Because the sinking velocity of large
cryogenic gypsum crystals is high the chance to catch large crystals with horizontal transects
directly under the ice should be lower compared to small crystals (Fig. 7A). Accordingly,
significant amounts of large cryogenic gypsum crystals were mainly sampled from the 0 m
layer where they could be scraped off the underside of the ice (see station 66, Tab. 2). In
contrast, smaller cryogenic gypsum crystals sink at lower velocities (Fig. 7A).  Hence, the
large quantity of small-sized crystals retrieved in the deeper layers of station 66, and all layers
of station 32 and 45 likely were influenced by the accumulated gypsum release in this size-
fraction, whereas the rarer large crystals indicated the momentary release at these stations.
The extremely large crystals sampled at station 80 at 10 m depth probably indicated an on-
going flux event during rapid melting. According to our dissolution experiments, gypsum
dissolution within Arctic surface waters should only have a minor impact on the size
distribution of cryogenic gypsum crystals within the surface water. Besides vertical flux,
advection of gypsum crystals with surface currents may also have influenced the size-
distribution of gypsum crystals sampled in the water column.

**4.2      Reasons why cryogenic gypsum was rarely observed in past studies**

The small temperature range of the -6.2 to -8.5 °C window, which is also the only gypsum
precipitation temperature spectrum applicable in the Arctic Ocean, has been considered one
reason why gypsum was not detected in other studies (Butler and Kennedy, 2015; Wollenburg
et al., 2018a). Furthermore, the kinetics of gypsum precipitation was considered as too slow
for detection during experimental studies, and the amount of gypsum hard to verify versus



other sea ice precipitates that are quantitatively much more abundant, leading the focus
towards other sea ice precipitates (Butler and Kennedy, 2015; Geilfus et al., 2013). Although
cryogenic mirabilite and hydrohalite are three and twenty-two times more abundant than
gypsum, respectively (Butler and Kennedy, 2015), gypsum is the only sea ice precipitate that
survives for one to several days within the Arctic water column. Cryogenic gypsum
dissolution increases with increasing hydrostatic pressure and increasing temperatures (Fig.
6). However, well preserved cryogenic gypsum crystals were retrieved from algae aggregates
collected from 2,146 m water depth, suggesting that either the transport from the surface to
this depth was very rapid or that dissolution was decreased and/or prevented once gypsum
crystals were included within the matrix of organosulfur compound-rich aggregates
(Wollenburg et al., 2018a). Yet, as seawater is usually undersaturated with respect to gypsum
(Briskin and Schreiber, 1978a; Briskin and Schreiber, 1978b) and is shown by our dissolution
experiments, disaggregation of organic aggregates would expose the gypsum to the seawater
and dissolve any crystals making it to the deep ocean or seafloor likely within a few days. The
same dissolution would occur within the sampling cups of sediment traps, explaining why
gypsum has not been observed in those type of samples.
Our dissolution experiments showed that cryogenic gypsum can persist long enough in the
cold polar surface water to be collected in measurable concentrations. The missing evidence
of gypsum from past studies was likely due to the quick dissolution of gypsum crystals at
higher temperatures and pressure dependence of dissolution kinetics, impeding the discovery
of gypsum in sediment trap samples and on the sea-floor. In addition, Formaldehyde
preservation leads to the immediate dissolution of gypsum, too, destroying any evidence of
cryogenic gypsum in all kinds of biological samples including water column and net samples.

**4.3    Potential of cryogenic gypsum as a ballast of algae blooms**

We found less than 6% dissolution of individual crystals in Polar Surface Water (PSW) per
day. Thus, at depths immediately below the fluorescence maximum where a significant part
of organic aggregates are formed (Iversen et al. 2010), the gypsum scavenging and ballasting
of aggregates (Turner, 2015) is little affected by gypsum dissolution (Olli et al., 2007) (Fig. 6,
Tab. 3). Incorporation of dense minerals into settling organic aggregates will increase their
density and, therefore, the size-specific sinking velocities of the aggregates (Iversen and
Ploug, 2010; Iversen and Robert, 2015; van der Jagt et al., 2018). The high sinking velocity of
large gypsum crystals >1 mm (5,000-7,000 m d$^{-1}$ (Fig. 7A)) could create strong hydrodynamic



shear that might cause disaggregation of fragile algae aggregates (Olli et al., 2007). This
supports previous suggestions of gypsum as an important ballast mineral of organic
aggregates, such as *Phaeocystis* (Wollenburg et al., 2018a) by rather small crystals.
As chlorophyll concentrations in the surface water were mostly low (< 1 mg m$^3$, H.F.
unpublished data), a massive gypsum-mediated export of phytoplankton was unlikely during
expedition PS106. However, especially at the ice floe of station 32/80, we observed a high
coverage of the ice underside by the filamentous algae *Melosira arctica*, and gypsum crystals
were found in *M. arctica* filaments collected nearby (Figs. 2D, 8). This indicates a potential
for rapid *M. arctica* downfall mediated by cryogenic gypsum, as soon as the algal filaments
were released from the melting sea ice. Hence, ballasting by cryogenic gypsum may also have
contributed to the mass export of *Melosira arctica* aggregates observed in 2012 (Boetius et al.

528    2013).



**5       Conclusions**
This study shows for the first time that gypsum released to the water at the onset of melt
season in the Arctic Ocean causes a constant flux of gypsum over wide spread areas and over
a long period of time (> six weeks). The morphological diversity of gypsum crystals retrieved
from Arctic surface waters and ice-cores indicated a complex variety of precipitation and
release processes as well as modifications during sea ice formation, the melt phase, and in the
water column. In the fresh and cold Polar surface water, gypsum crystals persist long enough
to act as an effective ballast on organic matter, such as phytoplankton filaments and marine
snow.

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

**Table captions:**






| Site | Date | Latitude (Deg N) | Longitude (Deg E) | Ocean depth (m) | Sampling depth | Water temp. (°C) | Salinity | Mean ice thickness (m) | Filtered water volume (m3) |
|------|------|------------------|-------------------|-----------------|----------------|------------------|----------|------------------------|----------------------------|
| 32 | 2017-06-15 | 81.73 | 10.86 | 1608 | under-ice | -1.94 | 34.27 | 1.90 | 2.2 |
|  |  |  |  |  | 5 m | n.a | n.a. | 1.90 | 3.9 |
| 45 | 2017-06-25 | 78.10 | 30.47 | 233 | under-ice | -1.52 | 33.84 | 1.00 | 2.3 |
|  |  |  |  |  | 5 m | -1.47 | 34.11 | 1.00 | 4.5 |
|  |  |  |  |  | 10 m | -1.68 | 34.29 | 1.00 | 2.5 |
| 66 | 2017-07-02 | 81.66 | 32.34 | 1506 | under-ice | -1.67 | 33.18 | 1.80 | 3.1 |
|  |  |  |  |  | 5 m | -1.71 | 33.76 | 1.80 | 2.7 |
|  |  |  |  |  | 10 m | -1.73 | 33.78 | 1.80 | 3.1 |
| 80 | 2017-07-12 | 81.37 | 17.13 | 1010 | 10 m | -1.37 | 32.87 | 1.80 | 1.7 |


Tab. 1: Properties of sea ice stations and characteristics of ROVnet profiles.



| Cruise, Site, mean water depth of the catch | >63 µm fraction | | | >30<63 µm fraction | | | >63 µm fraction weight% | >30<63 µm fraction weight% |
|---|---|---|---|---|---|---|---|---|
| | Mean length µm | Mean width µm | length/width ratio | Mean length µm | Mean width µm | length/width ratio | | |
| PS106.1, Stat. 32, 0 m | 68.46 | 44.27 | 1.55 | 50.64 | 35.03 | 1.45 | 43.70 | 56.30 |
| PS106.1, Stat. 32, 5 m | 63.28 | 35.90 | 1.76 | 49.91 | 35.57 | 1.40 | 33.72 | 66.28 |
| PS106.1, Stat. 32, mean (0-5 m) | 65.87 | 40.09 | 1.64 | 50.28 | 35.30 | 1.42 | 38.71 | 61.29 |
| PS106.2, Stat. 45, 0 m | 114.18 | 65.93 | 1.73 | 58.74 | 42.84 | 1.37 | 79.90 | 20.10 |
| PS106.2, Stat. 45, 5 m | 110.98 | 64.84 | 1.71 | 56.73 | 38.89 | 1.46 | 73.39 | 26.61 |
| PS106.2, Stat. 45 , 10 m | 92.83 | 46.81 | 1.98 | 50.32 | 29.98 | 1.68 | 66.14 | 33.86 |
| PS106.2, Stat. 45, mean (0-10 m) | 85.49 | 44.45 | 1.92 | 77.93 | 24.28 | 3.21 | 73.14 | 26.86 |
| PS106.2, Stat. 66, 0 m | 1355.38 | 415.10 | 3.27 | 56.67 | 25.63 | 2.21 | 99.25 | 0.75 |
| PS106.2, Stat. 66, 5 m | 411.42 | 73.45 | 5.60 | 62.03 | 12.20 | 5.08 | 75.23 | 24.77 |
| PS106.2, Stat. 66, 10 m | 101.40 | 23.19 | 4.37 | 30.71 | 5.79 | 5.30 | 61.18 | 38.82 |
| PS106.2, Stat. 66, mean (0-10 m) | 599.17 | 164.78 | 3.64 | 59.96 | 12.61 | 4.76 | 58.16 | 41.84 |
| PS106.2, Stat. 80, 10 m | 3078.44 | 1830.00 | 1.68 | 71.78 | 30.76 | 2.33 | 89.05 | 10.95 |


Tab. 2: Size measurements and percentage of mass contribution of gypsum crystals from the
>63 µm size fraction and the >30 < 63 µm size fraction

| | Dissolution in weight% | | | | |
|---|---|---|---|---|---|
| Chamber (no.)/Water mass | PSW | AW | EADW (1) | EADW (2) | EADW (3) |
| 1 | 11.34 | 76.22 | 47.52 | 57.08 | 74.92 |
| 2 | 1.33 | 86.23 | 26.09 | 71.03 | 53.77 |
| 3 | 8.29 | 82.93 | 21.05 | 47.15 | 57.43 |
| 4 | 2.99 | 78.57 | 10.91 | 58.56 | |
| **Mean** | **5.99** | **80.77** | **26.39** | **58.34** | **62.04** |


Tab. 3: Dissolution experiments on cryogenic gypsum crystals. 'Water mass' simulating
experiments with 34.9‰ sterile filtered seawater. Each experiment was conducted in parallel
in 3-4 separate pressure chambers.

**Figure captions:**


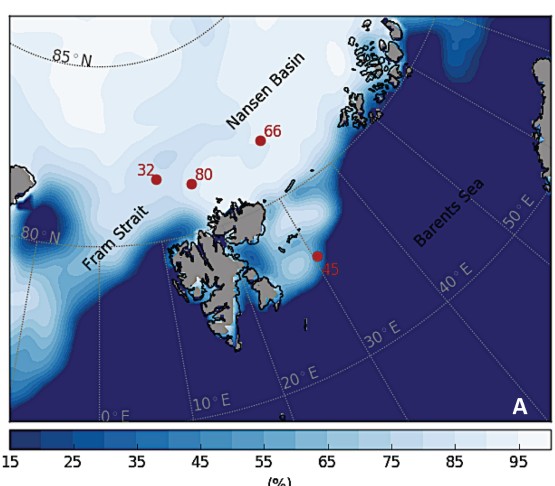

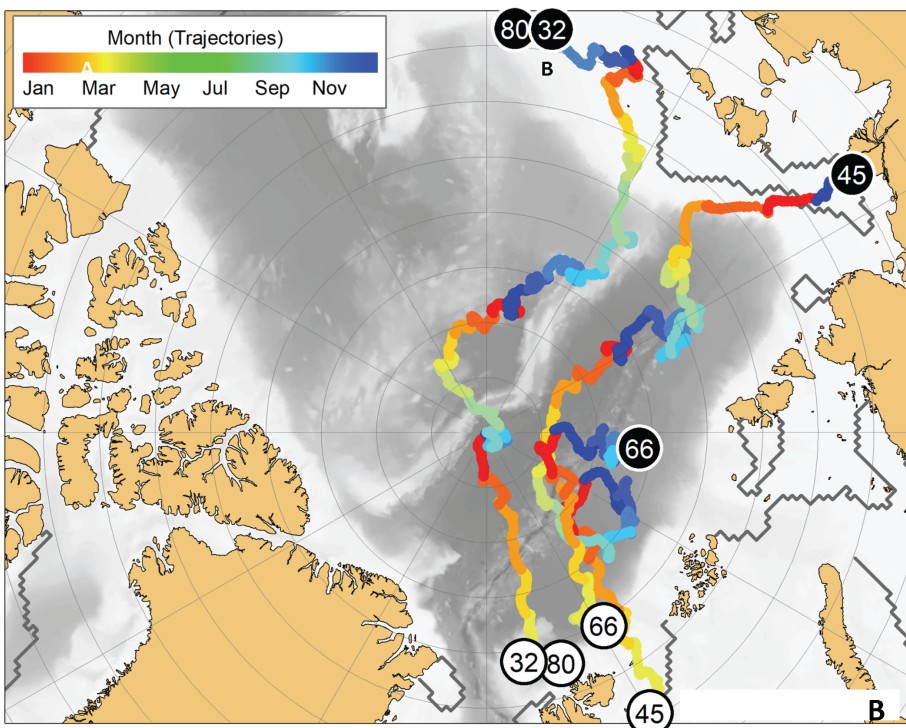


Fig. 1: Study area with sample locations. A: Sea ice coverage at the station and time of

sampling in %. B) Trajectories of the sea ice from which the cryogenic gypsum was released.

Each trajectory starts where sea ice formed (black circles), and shows its drift until the time


and place of sampling (white circles). The colour scale of the drift trajectories indicates the
month in which the back-tracked sea ice was at any given position.

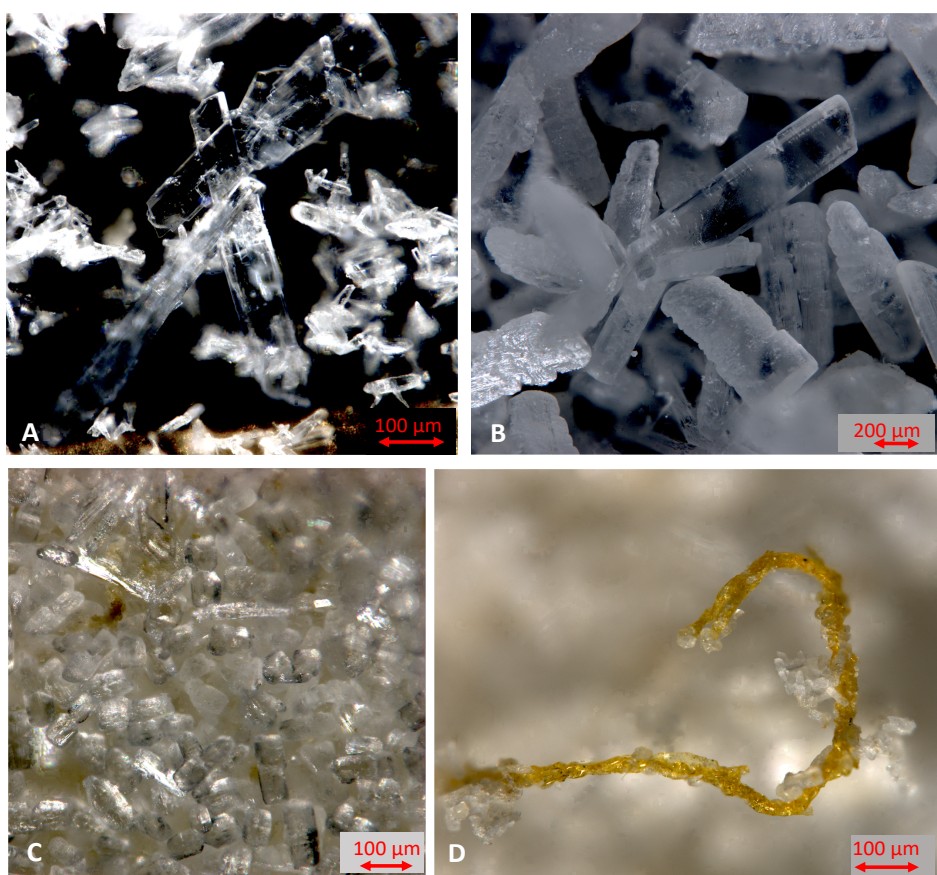


Fig. 2: Cryogenic gypsum crystals collected during Polarstern expedition PS106-1 from the
upper water column. A) Crystals collected from station 66 at 5 m water depth. B) Crystals
collected from station 66 at 0 m water depths. C) Crystals collected from station 45 at 10 m
water depth. D) Crystals collected from station 45 at 10 m water depths entangled in an algae
filament.







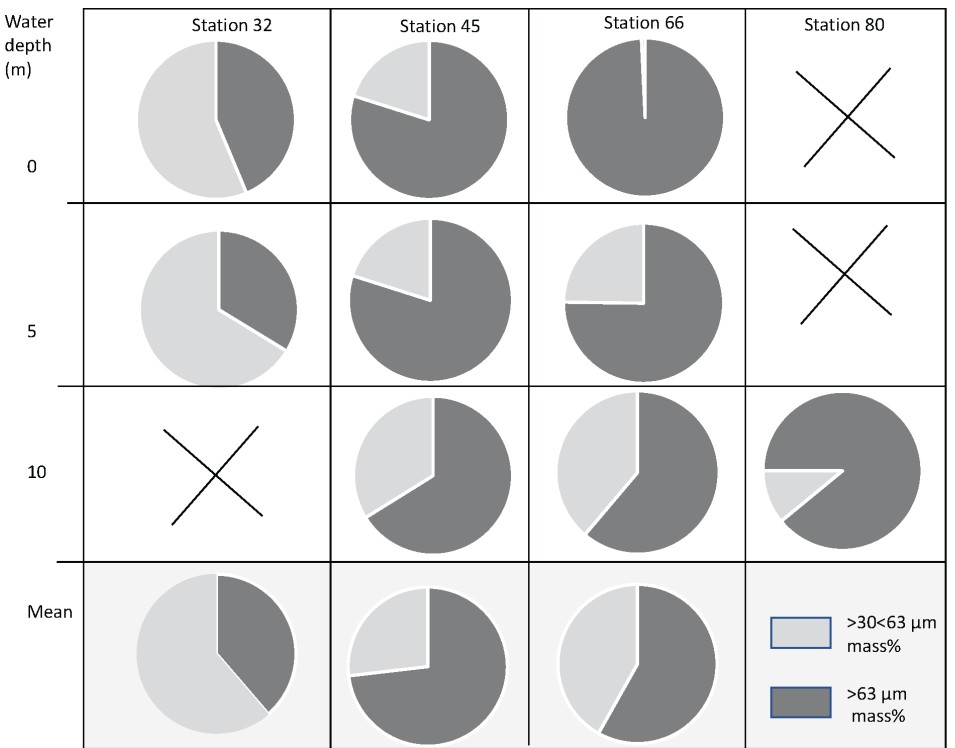


Fig. 3: Proportional mass (%) of cryogenic gypsum for the size fractions >30<63 $\mu$m and >63
$\mu$m for all ROV samples.


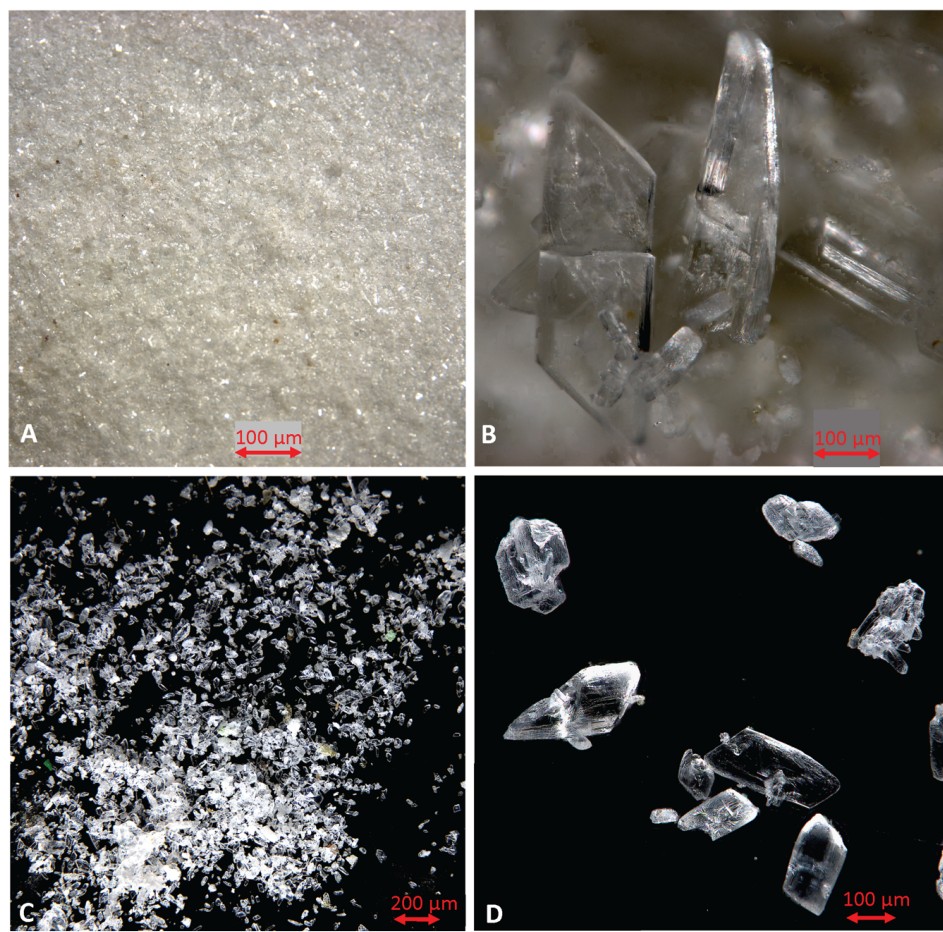

Fig. 4: Comparison of cryogenic gypsum crystals collected from the water column at station
PS45 (10 m water depth) (A-B) with crystals retrieved from an ice-core collected above the
ROVnet sampling area (C-D).



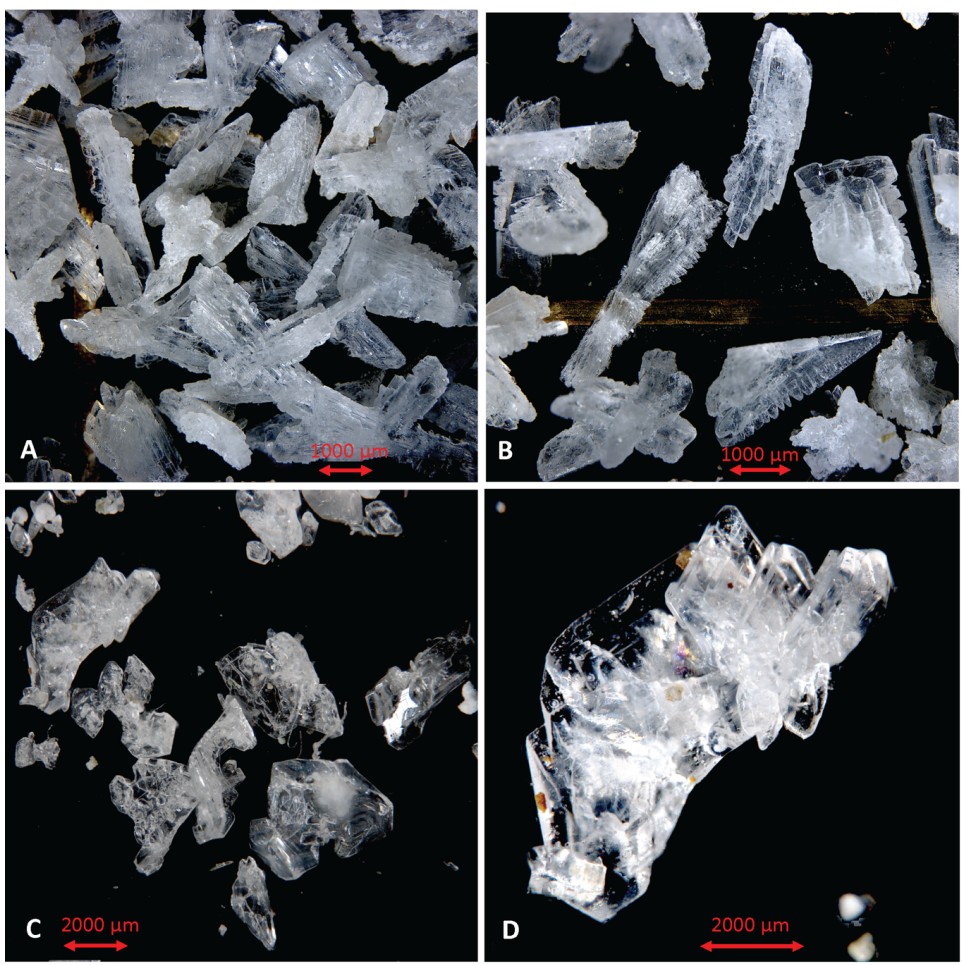


Fig. 5: Comparison of cryogenic gypsum crystals collected from the water column at station
PS80-2 (10 m water depth) (A-B) with crystals retrieved from an ice-core collected above the
ROVnet sampling area (C-D).


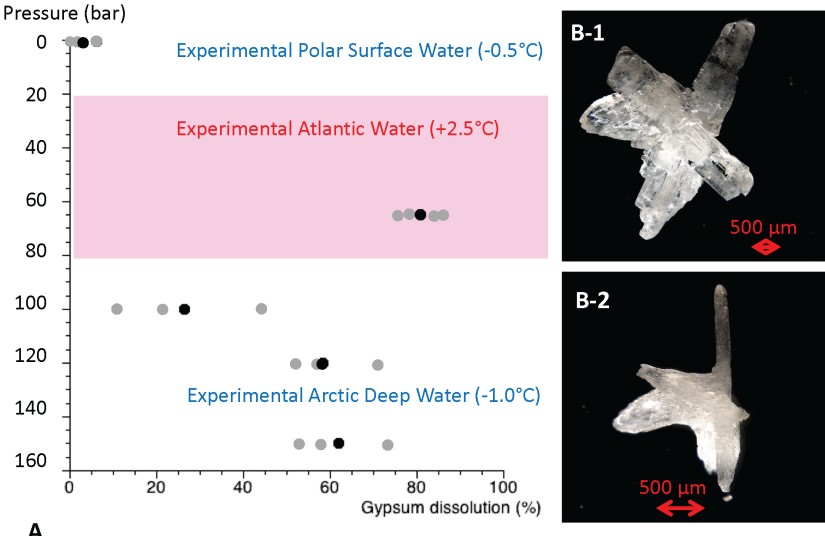

Figure 6: Results from cryogenic gypsum dissolution experiments. A) Graph showing the position of the simulated Arctic water masses in respect to pressure and temperature and how much gypsum (%) was dissolved on average over a 24-hours lasting exposure to such pressure and temperature conditions. Grey dots indicate the values from each aquarium, black dots the mean per experiment. B-1) Cryogenic gypsum crystal of the 120 bar-experiment before exposure. B-2) The same cryogenic gypsum crystal of the 120 bar-experiment after 24 hours.


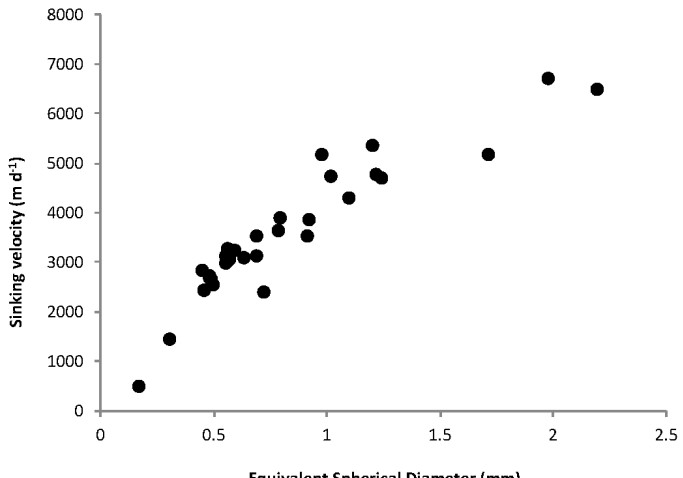

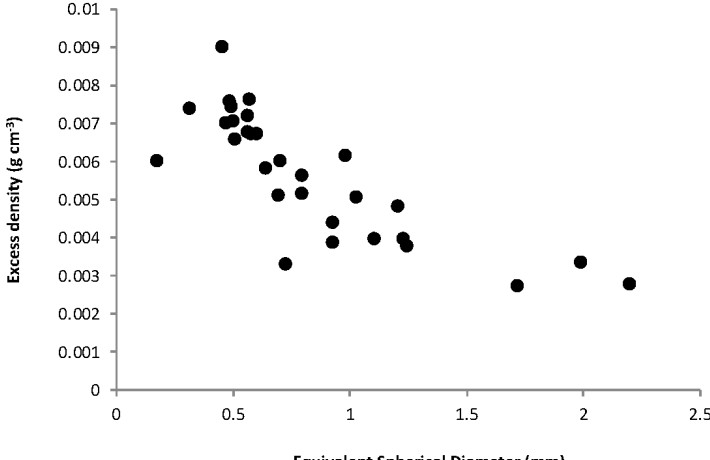


Fig. 7: (A) sinking velocity and (B) excess density (excess density = gypsum density –
seawater density) of cryogenic gypsum crystals plotted against equivalent spherical diameter
(ESD).

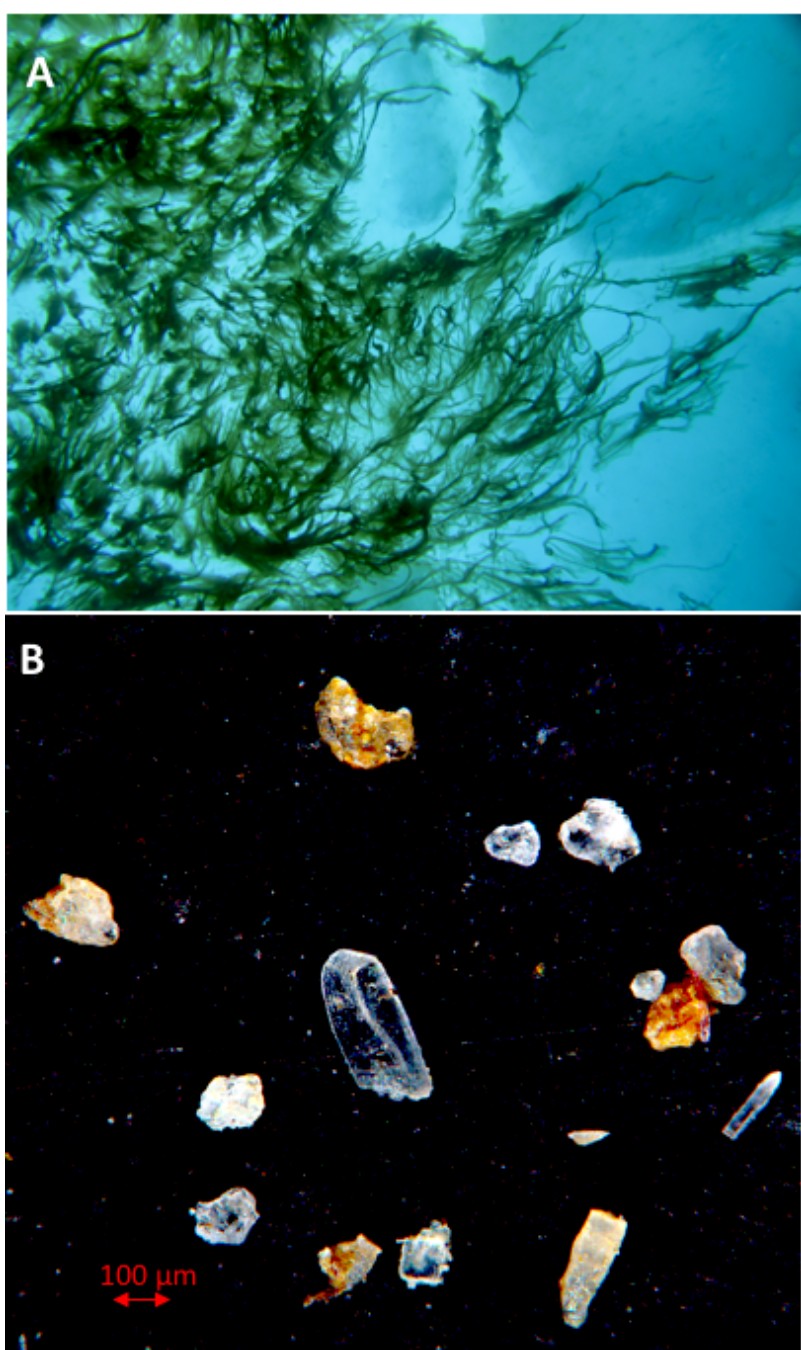


Fig. 8: Living *Melosira arctica* curtains hanging from ice flows during the PS106 expedition

(photo taken by M. Nicolaus and C. Katlein). Cryogenic gypsum isolated from *Melosira*

*arctica* (PS106-1, station 21(Peeken, 2018)).





**Acknowledgement:**
We thank Gernot Nehrke for performing Raman Spectroscopy on crystals from all catches.
Christoph Vogt and Dieter Wolf-Gladrow made valuable comments on the manuscript and we
thank them very much for it. We thank the captain and crew of RV Polarstern expedition
PS106 for their support at sea. This study was funded by the PACES (Polar Regions and
Coasts in a Changing Earth System) Program of the Helmholtz Association, the Helmholtz
Infrastructure Fund "Frontiers in Arctic Marine Monitoring (FRAM)". This study used
samples and data provided by the Alfred-Wegener-Institut Helmholtz-Zentrum für Polar- und
Meeresforschung in Bremerhaven from *Polarstern* expedition PS 106 (Grant No. AWI-
PS106_00).

**Author Contributions:**
J.W. lead the writing of this manuscript as well as gypsum sample preparation and analysis.
H.F., I.P., C.K., G.C., M.N. acquired ROVnet and ice samples in the field. M.I. measured
crystal settling velocities. T.K. performed the backtracking analysis. All authors contributed
to the writing and editing of the manuscript



