# Peer review of "New observations of the distribution, morphology, and dissolution dynamics of"

_The Cryosphere, 2019_

## Referee Comment (RC1) · Griet NEUKERMANS (Referee) · 30 Dec 2019

Review paper "New Observations of the distribution, morphology, and dissolution of cryogenic gypsum in the Arctic Ocean" by Wollenburg et al.

This paper provides a first description of the morphology, size, sinking speed, and dissolution of cryogenic gypsum crystals sampled in and under Arctic pack ice at four stations. These high-density crystals, precipitated during sea ice formation and released during ice melt, may potentially act as ballast mineral for organic material but have rarely been observed using traditional sampling methods due to difficulties of crystal preservation in samples (e.g. immediate dissolution in formaldehyde). Here, targeted

crystal sampling was carried out for the first time in situ, in the bottom of sea ice, and with a plankton net on a ROV at 0m and 6m below the sea ice. A detailed description of the sampled crystals is reported in this paper, including their morphology and size. Additional laboratory experiments were carried out to determine the sinking speed and dissolution conditions of the crystals. This work thus provides a first descriptive and quantitative sampling effort for cryogenic gypsum in the Arctic Ocean and I recommend publication of this work, but I have a number of major and minor comments that may help improve the paper.

Major comments:

This comment pertains to the potential ballast effect of cryogenic gypsum. The main motivation of this work is the potential ballast effect of cryogenic gypsum to help the sinking of organic matter through the Arctic water column. However, the excess density of cryogenic gypsum crystals derived from Equation 1 are very low, in the range 0.003-0.009 g cm-3. These values are in fact orders of magnitude lower than one would expect from the density of cryogenic gypsum; excess density = gypsum density – water density = 1.28 g cm-3. First, I think the reasons for the discrepancy between the expected and observed values should be better addressed. Second, an uncertainty assessment should be made on the measurements of particle diameter, sinking speed, and excess density. Third, how do these very low values of excess density of cryogenic crystals compare to the values of excess density of organic material? And are the excess density values of cryogenic gypsum high enough to provoke a ballast effect at all? Forth, a mineral material can only have a ballast effect on organic material if somehow the mineral gets associated with the organic material. Measuring sinking speed and density of a mineral alone does not prove its ballast effect. By which mechanisms could cryogenic gypsum get associated with organic material?

This comment pertains to the hypothesized link between sea ice texture/porosity and cryogenic gypsum crystal size/morphology (section 4.1). This is certainly a very interesting hypothesis, but it is not clear at all how your results support it. This section

needs improved clarity and better wording. The highly speculative nature of this entire section is obvious from the numerous occurrences of the words "likely", "may", "possibly", etc. Clarity could for example be improved by adding two rows at the top of Figure 3, one with a description of crystals in the ice core and one with a description of sea ice texture/porosity.

This work indicates a few of the reasons why cryogenic gypsum crystals have not been observed previously in scientific sampling efforts. I think the paper would benefit from a section of recommendations for future sampling.

Minor comments:

L29: given the difficulty of showing association between mineral and organic material and the absence of this association in your results, I suggest you replace the word "indicated" by "suggested".

L83: "to" should be "too"

L90: remove "best"

L103: insert "and" after "column"

L104: crystals (plural)

L106: the qualification of cryogenic gypsum as a ballast mineral is not demonstrated in this work in my opinion, as no association between gypsum and organic material has been clearly shown.

L122: can you add a photo or sketch of the rovnet?

Section 2.2: given all the sample handlings, the probability of crystal break up must be high?

Section 2.5: not clear how sinking speed was measured; did you have 2 cameras spaced 30 cm apart at the bottom and the top of the cylinder? What is the measurement uncertainty?

Section 2.6: You mention three tracking approaches. How did you combine them? Did you somehow average three different trajectories?

L378: please report density in g cm-3 for consistency.

L409-413: What is the relevance of these sentences?

L524: Fig 2D refers to crystals collected at station 45, not 32/80.

Fig 1: impossible to see the difference in trajectories 45 and 66.

L789: with respect to

Fig 7: please add your fitted curves to the figures.

---

## Author Comment (AC1) · 28 Jan 2020

Response to interactive comment of Griet Neukermans

Please find our response below the comment of Griet Neukermans

This paper provides a first description of the morphology, size, sinking speed, and dissolution of cryogenic gypsum crystals sampled in and under Arctic pack ice at four stations. These high-density crystals, precipitated during sea ice formation and released during ice melt, may potentially act as ballast mineral for organic material but have rarely been observed using traditional sampling methods due to difficulties of crystal

[Figure]

preservation in samples (e.g. immediate dissolution in formaldehyde). Here, targeted crystal sampling was carried out for the first time in situ, in the bottom of sea ice, and with a plankton net on a ROV at 0m and 6m below the sea ice.

Correction: Sampling was carried out at 0, 5 and 10 m below the ice. See section Materials and Methods, line 133.

A detailed description of the sampled crystals is reported in this paper, including their morphology and size. Additional laboratory experiments were carried out to determine the sinking speed and dissolution conditions of the crystals. This work thus provides a first descriptive and quantitative sampling effort for cryogenic gypsum in the Arctic Ocean and I recommend publication of this work, but I have a number of major and minor comments that may help improve the paper.

Major comments:

1) This comment pertains to the potential ballast effect of cryogenic gypsum. The main motivation of this work is the potential ballast effect of cryogenic gypsum to help the sinking of organic matter through the Arctic water column. However, the excess density of cryogenic gypsum crystals derived from Equation 1 are very low, in the range 0.003-0.009 g cm-3. These values are in fact orders of magnitude lower than one would expect from the density of cryogenic gypsum; excess density = gypsum density– water density = 1.28 g cm-3. First, I think the reasons for the discrepancy between the expected and observed values should be better addressed. Second, an uncertainty assessment should be made on the measurements of particle diameter, sinking speed, and excess density. Third, how do these very low values of excess density of cryogenic crystals compare to the values of excess density of organic material? And are the excess density values of cryogenic gypsum high enough to provoke a ballast effect at all?

R1)These are very good points and we thank the reviewer for pointing them out. The main reason for the discrepancies between the measured and the expected excess

density of gypsum is most likely due to the assumptions made for the size and volume estimates. We measured the size by converting the projected two-dimensional area of each gypsum crystal into an equivalent spherical diameter. We used the equivalent spherical diameter to calculate the volume of each gypsum crystal, assuming them to be solid gypsum. Here taking into account that the large crystals were inter-linking cylinders forming the shape of a cross or a star, we may have overestimated the gypsum volume and size. We also assumed the gypsum to be solid, but did see evidence of pores in the larger crystals, suggesting the crystals were porous. As we did the calculations for the manuscript we were also surprised about the low excess densities and calculated what the porosity of the gypsum crystals would have to be in order to get the density of the gypsum to ~2.3 g cm-3. By assuming a porosity of 0.5 we would obtain an average density of all the measured crystal of 2.22 ± 0.42 g cm-3. The issues with the porosity are mentioned in the material and methods in the section "Sinking velocities of gypsum crystal" in the last paragraph. We decided to be conservative and use the density based on the assumptions that the gypsum crystals were solid spheres since this is the most conservative estimates of the ballasting effect of gypsum. Organic matter without any ballast minerals is suspended and do not sink, so the addition of ballast would increase the sinking velocities of organic aggregates significantly. We have addressed the porosity issues in '4.1 Distribution and morphology of cryogenic gypsum crystals- line 408 but will clarify the porosity issues more intensely in the revised manuscript text under results.

2) Forth, a mineral material can only have a ballast effect on organic material if somehow the mineral gets associated with the organic material. Measuring sinking speed and density of a mineral alone does not prove its ballast effect. By which mechanisms could cryogenic gypsum get associated with organic material?

R2) When sea ice warms cryogenic gypsum can be released from widening brine channels and then fall directly in any organic material accumulating under the ice. This can be organic substances like exopolymer particels or alga. That way in Wollenburg et

al. (2018) cryogenic gypsum was found to amount to >50% of collected Phaeocystis aggregates sinking from a prevailing Phaeocystis bloom. During PS106 no strong bloom was observed, but even the isolated Melosira alga collected from under the ice during this expedition showed entrained gypsum crystals. Comprehensive studies on this topic are scheduled for the coming months/expeditions. So far unpublished student tests with settling cylinders corroborate a significant ballasting effect of gypsum for cultured Thalassiosira alga.

3) This comment pertains to the hypothesized link between sea ice texture/porosity and cryogenic gypsum crystal size/morphology (section 4.1). This is certainly a very interesting hypothesis, but it is not clear at all how your results support it. This section needs improved clarity and better wording. The highly speculative nature of this entire section is obvious from the numerous occurrences of the words "likely", "may", "possibly", etc.

R3) We agree with the reviewer that this section could be more precise, however when comparing cryogenic gypsum crystals sampled from the water column with crystals melted from ice cores that had been stored for > 1 year at -20°C the wording has to be cautious, since we do not know yet, how any storage or temperature change will affect the crystals. Thus we feel it is more honest to keep any conclusion in comparing those results rather speculative.

4) Clarity could for example be improved by adding two rows at the top of Figure 3, one with a description of crystals in the ice core and one with a description of sea ice texture/porosity.

R4) Figure 3 relates to ROV net samples only. Storing the ice core for several months and melting the ice to obtain the gypsum crystals may, as has been discussed in the manuscript, have slightly altered the original crystal size/shape.

5) This work indicates a few of the reasons why cryogenic gypsum crystals have not been observed previously in scientific sampling efforts. I think the paper would benefit

from a section of recommendations for future sampling.

R5) We will provide such recommendations in the revised version.

6) Minor comments: L29: given the difficulty of showing association between mineral and organic material and the absence of this association in your results, I suggest you replace the word "indicated" by "suggested".

R6) We will change this passage accordingly

7) L83: "to" should be "too"

R7) We will change this passage accordingly

8) L90: remove "best"

R8) We will change this passage accordingly

9) L103: insert "and" after "column"

R9) We will change this passage accordingly

10) L104: crystals (plural)

R10) We will change this passage accordingly

11) L106: the qualification of cryogenic gypsum as a ballast mineral is not demonstrated in this work in my opinion, as no association between gypsum and organic material has been clearly shown.

R11) As stated above, we found gypsum crystals adhering to Melosira alga, and in Wollenburg et al. (2018) comprising 50% of Phaeocystis aggregates, thus, gypsum can be incorporated in the two dominant Arctic alga and even in living Melosira. The experiments addressed the preservation potential and the sinking speed of single crystals of different shapes and sizes. The main aim here was to proof that gypsum crystals can actually sink to depth before being dissolved, which is crucial to proof its potential as ballast mineral.

12) L122: can you add a photo or sketch of the rov net?

R12) We will add such a photo in the revised version.

13) Section 2.2: given all the sample handlings, the probability of crystal break up must be high?

R13) Not with the ROV net samples. The first author is a micropaleontologist that handled the samples with utmost care, and as almost no alga or plankton was observed in the samples, the actual sieving was very moderate.

14) Section 2.5: not clear how sinking speed was measured; did you have 2 cameras spaced 30 cm apart at the bottom and the top of the cylinder? What is the measure-C3ment uncertainty?

R14) The measurements were only done with one camera, so a two-dimensional view. We measured over a distance of ∼5 cm after the crystals had reached terminal settling velocity and at stable and constant temperature and salinity. The technical uncertainties of the setup were smaller that the uncertainties between two similar sized gypsum crystals, which had up to 1000 m/d uncertainties (see figure 6 in the manuscript for crystals with equivalent spherical diameters of ∼1 mm. We will add this respective information in the revised manuscript version

15) Section 2.6: You mention three tracking approaches. How did you combine them? Did you somehow average three different trajectories?

R15) A weighted approach is used to determine the motion product that is applied for the tracking: The algorithm first checks for the availability of CERSAT motion data within a predefined search range. CERSAT provides the most consistent time series of motion vectors starting from 1991 to present and has shown good performance on the Siberian shelves. During summer months (June–August) when drift estimates from CERSAT are missing, motion information is bridged with OSISAF (2012 to present). If no valid OSISAF motion vector is available within the search range, NSIDC

data is applied. A detailed method description is provided in Krumpen et al. 2019 (https://doi.org/10.1038/s41598-019-41456-y)

16) L378: please report density in g cm-3 for consistency.

R16) We will include this in the manuscript – the density was 2.22 ±0.42 g cmˆ-3.

17) L409-413: What is the relevance of these sentences?

R17) It is just a description of what has been observed.

18) L524: Fig 2D refers to crystals collected at station 45, not 32/80.

R18) Correct. Fig. 8 refers to being close to stations 32/80. We will change the text to 'However, especially at the ice floe of station 32/80, we observed a high coverage of the ice underside by the filamentous algae Melosira arctica, and gypsum crystals were found in M. arctica filaments collected nearby (Fig. 8) as well as at station 45 (Fig. 2D)' in the revised version.

19) Fig 1: impossible to see the difference in trajectories 45 and 66.

R19) The starting point of each trajectory is indicated by a black, the end point by a respective white label. The trajectories are indeed close together. However, they are distinguishable by the colour-coding. The tracking of both ice floes started in the same month (July). Since the ice floe of Station 45 had a longer trajectory than Station 65, it had passed the position of Station 65 in March, which is why its trajectory was plotted in orange in that region. Following the colour scale of Figure 1 backwards, we can see that floe 45 made a circular turn in the Nansen Basin in winter 2016/17, but had actually been a 2-year floe probably originating from the Laptev Sea, whereas floe 66 probably formed in autumn 2016 in the Nansen Basin.

20) L789: with respect to Fig 7: please add your fitted curves to the figures.

R20) Regression lines will be added to the curves in the revised manuscript

---

## Referee Comment (RC2) · Anonymous Referee #2 · 11 Feb 2020

This is an excellently executed investigation on a subject that long has been around in the vertical flux literature. It is clearly and well written. I am not a chemical oceanographer and can thus not evaluate most of the chemical analyses. However, the description of the morphology, size, dissolution nd sinking velocity of cryogenic gypsum particles is a major break-through for vertical flux regulation in ice-covered waters.

When it comes to the regulation of vertical export of biogenic particles in the Arctic Ocean, in particular the sinking and non-sinking of phytoplankton and ice algae this manuscripts provides mechanisms that are of great interest. I would have liked to see some speculation in this direction. To the candidates that have been discussed

previously belong Phaeocystis with sinks (1, 2) or does not (3). Similar speculations also exist for Melosira arctica. The authors may have the mechanism to understand the pelagic-benthic coupling in the Artic Ocean in their hands. This deserves some high-thinking. How will for example warming of surface waters below sea ice impact the sinking of biogenic matter and bloom development in the future?

(1) Wassmann, P., Vernet, M., Mitchell, G., Rey, P. (1990). Mass sedimentation of Phaeocystis pouchetii in the Barents Sea during spring. Mar. Ecol. Prog. Ser. 66: 183-195.

(2) Hamm, C., M. Reigstad, C. Wexels Riser, A. Mühlebach & P. Wassmann (2001). On the trophic fate of Phaeocystis pouchetii: VII. Sedimentation of Phaeocystis-derived organic matter via krill fecal strings during a Phaeocystis bloom in the Balsfjord, northern Norway. Mar. Ecol. Prog. Ser. 209: 55-69.

(3) Reigstad, M., Wassmann, P. (2007). Does Phaeocystis spp. contribute significantly to vertical export of biogenic matter? Biogeochemistry 83 (1-3): 217-234

---

## Author Comment (AC2) · 18 Feb 2020

We are very grateful for the praising evaluation of our manuscript by anonymous reviewer 2. Apart from being positive about the manuscript reviewer 2 asks for some speculative sentences on our finding in context to published papers on the fate of Phaeocystis /Melosira in carbon export, and to changes in the future Arctic Ocean. These suggestions conflicts with the suggestions of reviewer 1 who requested to be less speculative. The question why a 'unsinkable alga' like Phaeocystis can sink once ballasted by cryogenic gypsum has also already been addressed by Wollenburg et al., 2018. We are continuing our research and will address these questions again with

numbers and facts rather than to be just speculative in our next paper. This manuscript was dedicated to what the titel says 'New observations of the distribution, morphology, and dissolution dynamics of cryogenic gypsum in the Arctic Ocean'

---

## Author Response (AR2)

**Please find our point per point responses to each comment in bold**

*Response to interactive comment of Griet Neukermans*

This paper provides a first description of the morphology, size, sinking speed, and dissolution of cryogenic gypsum crystals sampled in and under Arctic pack ice at four stations. These high-density crystals, precipitated during sea ice formation and released during ice melt, may potentially act as ballast mineral for organic material but have rarely been observed using traditional sampling methods due to difficulties of crystal preservation in samples (e.g. immediate dissolution in formaldehyde). Here, targeted crystal sampling was carried out for the first time in situ, in the bottom of sea ice, and with a plankton net on a ROV at 0m and 6m below the sea ice.
**Correction: Sampling was carried out at 0, 5 and 10 m below the ice. See section Materials and Methods, line 133.**

A detailed description of the sampled crystals is reported in this paper, including their morphology and size. Additional laboratory experiments were carried out to determine the sinking speed and dissolution conditions of the crystals. This work thus provides a first descriptive and quantitative sampling effort for cryogenic gypsum in the Arctic Ocean and I recommend publication of this work, but I have a number of major and minor comments that may help improve the paper.

Major comments:

1) This comment pertains to the potential ballast effect of cryogenic gypsum. The main motivation of this work is the potential ballast effect of cryogenic gypsum to help the sinking of organic matter through the Arctic water column. However, the excess density of cryogenic gypsum crystals derived from Equation 1 are very low, in the range 0.003-0.009 g cm-3. These values are in fact orders of magnitude lower than one would expect from the density of cryogenic gypsum; excess density = gypsum density– water density = 1.28 g cm-3. First, I think the reasons for the discrepancy between the expected and observed values should be better addressed. Second, an uncertainty assessment should be made on the measurements of particle diameter, sinking speed, and excess density. Third, how do these very low values of excess density of cryogenic crystals compare to the values of excess density of organic material? And are the excess density values of cryogenic gypsum high enough to provoke a ballast effect at all?

**R1) These are very good points and we thank the reviewer for pointing them out. We have extensively discussed the issues raised on the excess density by you and reviewer 1 and decided to remove the plot with the excess density from the manuscript since it is just a calculation based on Stokes Law. We are confident that the actual measured settling velocities and crystal sizes are correct, however, Stokes Law is made for settling spheres with Reynolds Numbers much smaller than 1. Neither of these criteria are met for the gypsum crystals.**
**After going through the manuscript in great details, we are confident that by removing the excess density part of the story is not impacting the main conclusions or the supporting data to make those conclusions. The main point is that gypsum sinks very, very fast through the water column and will ballast settling aggregates if it is incorporated into the aggregates. This will increase the sinking velocities of the aggregates and provide a fast and stronger link between the surface ocean and the deep sea and seafloor. As it turned out, we have not used the excess densities in the discussion or the abstract, we were aware that**

these data were based on approximations since there is no alternative to Stokes Law, which strictly does not apply to gypsum crystals. Therefore, we removed the excess densities from the manuscript, without the need to change the discussion or abstract.

We did test for porosity of the gypsum crystals to see if that could explain the low excess densities. We did this via SEM analyses on the crystals (all sites and size classes). We did not find any evidence for strongly increased porosities but found that the crystals looked solid. The large crystals had more complex structures and seemed to have increased surface roughness, but not to the extent that would explain the low excess densities. We therefore conclude that the results for the excess densities was because the gypsum crystals had high Reynolds numbers – at times larger than 100 – and were not spherical. The use of Stokes Law to calculate excess density is generally limited to spherical, non-permeable objects with Reynolds Numbers much smaller than 1.

2) Forth, a mineral material can only have a ballast effect on organic material if somehow the mineral gets associated with the organic material. Measuring sinking speed and density of a mineral alone does not prove its ballast effect. By which mechanisms could cryogenic gypsum get associated with organic material?

R2) When sea ice warms cryogenic gypsum can be released from widening brine channels and then fall directly in any organic material accumulating under the ice. This can be organic substances like exopolymer particels or algae. That way in Wollenburg et al. (2018) cryogenic gypsum was found to amount to >50% of collected *Phaeocystis* aggregates sinking from a prevailing *Phaeocystis* bloom. During PS106 no strong bloom was observed, but even the isolated *Melosira* alga collected from under the ice during this expedition showed entrained gypsum crystals. Comprehensive studies on this topic are scheduled for the coming months/expeditions. So far unpublished student tests with settling cylinders corroborate a significant ballasting effect of gypsum for cultured *Thalassiosira* algae.

3) This comment pertains to the hypothesized link between sea ice texture/porosity and cryogenic gypsum crystal size/morphology (section 4.1). This is certainly a very interesting hypothesis, but it is not clear at all how your results support it. This section needs improved clarity and better wording. The highly speculative nature of this entire section is obvious from the numerous occurrences of the words "likely", "may", "possibly", etc.
R3) We agree with the reviewer that this section could be more precise, however when comparing cryogenic gypsum crystals sampled from the water column with crystals melted from ice cores that had been stored for > 1 year at -20°C the wording has to be cautious, since we do not know yet, how any storage or temperature change will affect the crystals. Thus, we feel it is more honest to keep any conclusion in comparing those results rather speculative.

4) Clarity could for example be improved by adding two rows at the top of Figure 3, one with a description of crystals in the ice core and one with a description of sea ice texture/porosity.
R4) Figure 3 relates to ROV net samples only. Storing the ice core for several months and melting the ice to obtain the gypsum crystals may, as has been discussed in the manuscript, have slightly altered the original crystal size/shape.

5) This work indicates a few of the reasons why cryogenic gypsum crystals have not been observed previously in scientific sampling efforts. I think the paper would benefit from a section of recommendations for future sampling.
**R5) A sampling protocol is provided as S9 of the supplementary file.**

6) Minor comments: L29: given the difficulty of showing association between mineral and organic material and the absence of this association in your results, I suggest you replace the word "indicated" by "suggested".
**R6) We have changed this passage accordingly.**

7) L83: "to" should be "too"
**R7) We have changed this passage accordingly.**

8) L90: remove "best"
**R8) We have changed this passage accordingly.**

9) L103: insert "and" after "column"
**R9) We have changed this passage accordingly.**

10) L104: crystals (plural)
**R10) We have changed this passage accordingly.**

11) L106: the qualification of cryogenic gypsum as a ballast mineral is not demonstrated in this work in my opinion, as no association between gypsum and organic material has been clearly shown.
**R11) As stated above, we found gypsum crystals adhering to *Melosira* alga, and in Wollenburg et al. (2018) comprising 50% of *Phaeocystis* aggregates, thus, gypsum can be incorporated in the two dominant Arctic alga and even in living *Melosira*. The experiments addressed the preservation potential and the sinking speed of single crystals of different shapes and sizes. The main aim here was to proof that gypsum crystals can actually sink to depth before being dissolved, which is crucial to proof its potential as ballast mineral.**

12) L122: can you add a photo or sketch of the rov net?
**R12) We have added such a photo as supplementary fig. 1 in the revised version.**

13) Section 2.2: given all the sample handlings, the probability of crystal break up must be high?
**R13) Not with the ROV net samples. The first author is a micropaleontologist that handled the samples with utmost care, and as almost no alga or plankton was observed in the samples, the actual sieving was very moderate.**

14) Section 2.5: not clear how sinking speed was measured; did you have 2 cameras spaced 30 cm apart at the bottom and the top of the cylinder? What is the measure-C3ment uncertainty?
**R14) The measurements were only done with one camera, so a two-dimensional view. We measured over a distance of ~5 cm after the crystals had reached terminal settling velocity and at stable and constant temperature and salinity. The technical uncertainties of the setup were smaller that the uncertainties between two similar sized gypsum crystals, which had up to 1000 m/d uncertainties (see figure 6 in the manuscript for crystals with equivalent spherical diameters of ~1 mm). We will add this respective information in the revised manuscript version**

15) Section 2.6: You mention three tracking approaches. How did you combine them? Did you somehow average three different trajectories?
**R15) As written in the manuscript, the reader is referred to Krumpen et al. 2019 (https://doi.org/10.1038/s41598-019-41456-y) for details on this approach.**

16) L378: please report density in g cm-3 for consistency.
**R16) We have changed this passage accordingly.**

17) L409-413: What is the relevance of these sentences?
**R17) It is just a description of what has been observed.**

18) L524: Fig 2D refers to crystals collected at station 45, not 32/80.
**R18) Correct. Fig. 8 refers to being close to stations 32/80. We will change the text to 'However, especially at the ice floe of station 32/80, we observed a high coverage of the ice underside by the filamentous algae *Melosira arctica*, and gypsum crystals were found in *M. arctica* filaments collected nearby (Fig. 8) as well as at station 45 (Fig. 2D)' from L499 onward in the revised version.**

19) Fig 1: impossible to see the difference in trajectories 45 and 66.
**R19) The starting point of each trajectory is indicated by a black, the end point by a respective white label. The trajectories are indeed close together. However, they are distinguishable by the colour-coding. The tracking of both ice floes started in the same month (July). Since the ice floe of Station 45 had a longer trajectory than Station 65, it had passed the position of Station 65 in March, which is why its trajectory was plotted in orange in that region. Following the colour scale of Figure 1 backwards, we can see that floe 45 made a circular turn in the Nansen Basin in winter 2016/17, but had actually been a 2-year floe probably originating from the Laptev Sea, whereas floe 66 probably formed in autumn 2016 in the Nansen Basin.**

20) L789: with respect to Fig 7: please add your fitted curves to the figures.
**R20) A regression line was added to the curves in the revised manuscript.**

*Response to Anonymous Referee #2*

This is an excellently executed investigation on a subject that long has been around in the vertical flux literature. It is clearly and well written. I am not a chemical oceanographer
and can thus not evaluate most of the chemical analyses. However, the description of the morphology, size, dissolution nd sinking velocity of cryogenic gypsum particles is a major break-through for vertical flux regulation in ice-covered waters. When it comes to the regulation of vertical export of biogenic particles in the Arctic Ocean, in particular the sinking and non-sinking of phytoplankton and ice algae this manuscripts provides mechanisms that are of great interest. I would have liked to see some speculation in this direction. To the candidates that have been discussed
previously belong Phaeocystis with sinks (1, 2) or does not (3). Similar speculations also exist for Melosira arctica. The authors may have the mechanism to understand the pelagic-benthic coupling in the Artic Ocean in their hands. This deserves some high-thinking. How will for example warming of surface waters below sea ice impact the sinking of biogenic matter and bloom development in the future?
(1) Wassmann, P., Vernet, M., Mitchell, G., Rey, P. (1990). Mass sedimentation of Phaeocystis pouchetii in the Barents Sea during spring. Mar. Ecol. Prog. Ser. 66: 183-195.
(2) Hamm, C., M. Reigstad, C. Wexels Riser, A. Mühlebach & P. Wassmann (2001). On the trophic fate of Phaeocystis pouchetii: VII. Sedimentation of Phaeocystis-derived organic
matter via krill fecal strings during a Phaeocystis bloom in the Balsfjord, northern Norway. Mar. Ecol. Prog. Ser. 209: 55-69.
(3) Reigstad, M., Wassmann, P. (2007). Does Phaeocystis spp. contribute significantly to vertical export of biogenic matter? Biogeochemistry 83 (1-3): 217-234

**Response to interactive comment by reviewer 2**

**Apart from being positive about the manuscript reviewer 2 asks for some speculative sentences on our finding in context to published papers on the fate of *Phaeocystis /Melosira* in carbon export, and to changes in the future Arctic Ocean. These suggestions conflict with the suggestions of reviewer 1 who requested to be less speculative. Furthermore, the question why a 'unsinkable alga' like *Phaeocystis* can sink once ballasted by cryogenic gypsum has also already been addressed by Wollenburg et al., 2018. We are continuing our research and will address these questions again with numbers and facts rather than to be just speculative in our next paper. As this manuscript is dedicated to what the titel says 'New observations of the distribution, morphology, and dissolution dynamics of cryogenic gypsum in the Arctic Ocean' we restrained from speculative outlooks on the future carbon pump.**

**Response to Editor decision 19_2_2020**

The topic of your paper is timely and interesting. You provide original data which will be clearly beneficial to the community. The reviews are for the most part positive. I therefore encourage you to submit a revised version that I will review and possibly, but not necessarily, send again to reviewers. One issue that needs to be strengthened is the actual ballasting effect of the gypsum crystals. You morphological analysis clearly would benefit from a more elaborate aspect. For example, have you thought of measuring the specific surface area of the crystals to address the porosity issue? This would really be enlightening. An adequate method for this type of solid may be the BET method using methane adsorption. There is an appropriate system at PSI in Switzerland. Building your own system would probably be too time-consuming. I strongly suggest you explore this issue before submitting a revised version. I am ready to accept that such measurements cannot be done within the usual time frame for a revision, but please at least give it a try.

**We have extensively discussed the issues raised on the excess density by you and reviewer 1 and decided to remove the plot with the excess density from the manuscript since it is just a calculation based on Stokes Law. We are confident that the actual measured settling velocities and crystal sizes are correct, however, Stokes Law is made for settling spheres with Reynolds Numbers much smaller than 1. Neither of these criteria are met for the gypsum crystals.**
**After going through the manuscript in great details, we are confident that by removing the excess density part of the story is not impacting the main conclusions or the supporting data to make those conclusions. The main point is that gypsum sinks very, very fast through the water column and will ballast settling aggregates if it is incorporated into the aggregates. This will increase the sinking velocities of the aggregates and provide a fast and stronger link between the surface ocean and the deep sea and seafloor. As it turned out, we have not used the excess densities in the discussion or the abstract, we were aware that these data were based on approximations since there**

is no alternative to Stokes Law, which strictly does not apply to gypsum crystals. Therefore, we removed the excess densities from the manuscript, without the need to change the discussion or abstract.

We did test for porosity of the gypsum crystals to see if that could explain the low excess densities. We did this via SEM analyses on the crystals (all sites and size classes) and exemplified pictures are shown in S8 of the supplements of the revised manuscript. We did not find any evidence for strongly increased porosities but found that the crystals looked solid. The large crystals had more complex structures and seemed to have increased surface roughness, but not to the extent that would explain the low excess densities. We therefore conclude that the results for the excess densities was because the gypsum crystals had high Reynolds numbers – at times larger than 100 – and were not spherical. The use of Stokes Law to calculate excess density is generally limited to spherical, non-permeable objects with Reynolds Numbers much smaller than 1.

[revised manuscript text omitted]